# Predicting *Plasmodium falciparum* infection status in blood using a multiplexed bead-based antigen detection assay and machine learning approaches

Sarah E. Schmedes[1,2¤], Rafael P. Dimbu[3], Laura Steinhardt[1], Jean F. Lemoine[4], Michelle A. Chang[1], Mateusz Plucinski[1,5], Eric Rogier[1]*

1 Malaria Branch, Division of Parasitic Diseases and Malaria, Centers for Disease Control and Prevention, Atlanta, Georgia, United States of America, 2 Association of Public Health Laboratories, Silver Spring, Maryland, United States of America, 3 National Malaria Control Program, Ministry of Health, Luanda, Angola, 4 Programme National de la Contrôle de la Malaria, Ministère de la Santé Publique et de la Population, Port-au-Prince, Haiti, 5 U.S. President's Malaria Initiative, Centers for Disease Control and Prevention, Atlanta, GA, United States of America

¤ Current address: Bureau of Public Health Laboratories, Florida Department of Health, Jacksonville, Florida, United States of America

* erogier@cdc.gov

**Data Availability Statement:** As instituted by the local IRBs, anonymized data is available upon request from the national ministries of health for

## Abstract

### Background

*Plasmodium* blood-stage infections can be identified by assaying for protein products expressed by the parasites. While the binary result of an antigen test is sufficient for a clinical result, greater nuance can be gathered for malaria infection status based on quantitative and sensitive detection of *Plasmodium* antigens and machine learning analytical approaches.

### Methods

Three independent malaria studies performed in Angola and Haiti enrolled persons at health facilities and collected a blood sample. Presence and parasite density of *P. falciparum* infection was determined by microscopy for a study in Angola in 2015 (n = 193), by qRT-PCR for a 2016 study in Angola (n = 208), and by qPCR for a 2012–2013 Haiti study (n = 425). All samples also had bead-based detection and quantification of three *Plasmodium* antigens: pAldolase, pLDH, and HRP2. Decision trees and principal component analysis (PCA) were conducted in attempt to categorize *P. falciparum* parasitemia density status based on continuous antigen concentrations.

### Results

Conditional inference trees were trained using the known *P. falciparum* infection status and corresponding antigen concentrations, and PCR infection status was predicted with accuracies ranging from 73–96%, while level of parasite density was predicted with accuracies ranging from 59–72%. Multiple decision nodes were created for both pAldolase and HRP2

Angola (geral@minsa.gov.ao; +244 222 330 473) and Haiti (info@mspp.gouv.ht; (509)4432-8401). All R code for data analysis and figure generation is available at https://github.com/SESchmedes/plasmodium_falciparum_infection_status.

**Funding:** MP was supported by the U.S. President's Malaria Initiative. The funders had no role in study design, data collection and analysis, decision to publish, or preparation of the manuscript.

**Competing interests:** The authors have declared that no competing interests exist.

antigens. For all datasets, dichotomous infectious status was more accurately predicted when compared to categorization of different levels of parasite densities. PCA was able to account for a high level of variance (>80%), and distinct clustering was found in both dichotomous and categorical infection status.

## Conclusions

This pilot study offers a proof-of-principle of the utility of machine learning approaches to assess *P. falciparum* infection status based on continuous concentrations of multiple *Plasmodium* antigens.

## Introduction

Malaria remains a significant global public health burden that is responsible for an estimated 229 million infections worldwide and 409,000 deaths annually, with the vast majority of malaria cases and deaths occurring in Sub-Saharan Africa [1]. The knowledge of malaria epidemiology and implementation of control measures in any endemic setting are imperative for the reduction of transmission and eventual transition to elimination efforts [2]. Utilizing rapid diagnostic tests (RDTs), detection of *Plasmodium* antigens directly from human blood provides an effective measure of active malaria infection [3]. In 2019, 348 million RDTs were sold by global manufacturers with the most commonly used RDTs detecting the presence of *Plasmodium falciparum* histidine rich protein 2 (HRP2), though tests are available which also detect *Plasmodium* aldolase and lactate dehydrogenase (LDH) [1]. These RDTs are evaluated at a sensitivity of detection of 200 parasites/μL, although actual field results can be influenced by a number of test, operator, and parasite factors [4]. A positive HRP2-based RDT result could indicate an active *P. falciparum* infection (clinical or subpatent) or a recently-cleared *P. falciparum* infection with HRP2 antigen still in systemic circulation. Due to the length of time for post-treatment clearance of HRP2, HRP2-based RDTs can be positive for weeks after resolution of infection [5, 6]. Clearance of aldolase and LDH is substantially quicker and their presence is more indicative of active infection [7]. A negative result for any type of RDT can indicate a true negative, or false negative due to a low-density *Plasmodium* infection or low production of the antigen target (or non-production of HRP2 due to a gene deletion, [8]). RDTs provide a qualitative, point-of-care measurement for specific *Plasmodium* antigens (presence or absence of the antigen), and quantitative measurement of these malaria antigens can occur in the laboratory setting through different immunoassay platforms [9–11].

Further molecular testing, such as real-time polymerase chain reaction can be used as a sensitive method to detect the presence/absence of active infections for all malaria species and inform estimates of malaria prevalence [12]. However, testing is more laborious and costly as multiple steps and assays are needed for confirmation of *Plasmodium* DNA. Initial screening of samples using antigen detection can serve as a more economical and high-throughput method to screen samples to predict parasite presence/absence status using PCR. Additionally, beyond the simple presence or absence of a *Plasmodium* antigen in a sample, using lab assays for detection of multiple antigens can provide quantitative estimates for each target, as well as generate an antigen profile (interpretation for +/- to multiple targets) for a specimen [9].

Recent advances in statistics and computing power have seen the increase of use of sophisticated machine learning approaches for classification in the context of complex datasets [13, 14], including random forest machine learning approaches to predict protection to malaria

based on antibody profiles [15]. In this study, we evaluated the use of machine learning approaches using continuous concentration of antigen data to predict PCR presence/absence classification. Specifically, we evaluated the use of conditional inference trees using antigen concentration and log concentration to predict the presence/absence of infection and classification of five parasite density levels using dried blood spot samples from high and low transmission areas in Angola and Haiti. Predictive models can provide greater nuance to epidemiological estimates and inform the selection of samples as a screening method for further downstream molecular testing.

## Materials and methods

### Samples and ethics statement

Dried blood spot (DBS) samples used in this study were previously collected for a therapeutic efficacy study (TES) in Angola in 2015 (n = 193) [16], an Angolan health facility (HF) survey in 2016 (n = 208) [17], and a bednet study in Haiti enrolling persons seeking care in health facilities from 2012–2013 (n = 425) [18]. The TES samples were from symptomatic children seeking care at health facilities with microscopy confirmed *P. falciparum* infection. The 2016 health facility survey samples were from a representative sample of febrile and afebrile outpatients of all ages in Angola. The Angolan TES activity was classified as non-research by human subjects research boards at CDC (#2014-233b) and the Angolan Ministry of Health. Blood sample collection during the Angolan health facility survey was approved by the Angolan Ministry of Health and further laboratory investigation approved by the Office of the Associate Director for Science in the Center for Global Health at the CDC (#2018–034). The Haiti bednet study enrolled febrile patients presenting to health facilities and capillary blood was collected blood for an RDT and also spotted onto filter paper. The Haiti study protocol was approved by the National Bioethics Committee of Haiti and the Institutional Review Board at the CDC. For all studies, written consent was obtained from all participants, and assent obtained from minors upon consent from minor parent or guardian.

### *Plasmodium falciparum* detection and parasite density calculation for different studies

Molecular detection of *P. falciparum* infection and parasite density estimation were determined using real-time PCR and/or microscopy. Parasitemia for samples collected during the 2015 TES in Angola was determined using traditional microscopy [16]. Parasitemia for samples collected during the 2016 health facility survey in Angola was determined using sensitive quantitative PCR (sen-qPCR) using methods previously described [9], with an analytical sensitivity of 0.02 parasites/µL [19]. Parasitemia for samples collected in Haiti was determined using PET-PCR using methods previously described [18].

### PCR assays and multiplex antigen detection

For samples with PCR results, total DNA was extracted from blood specimens by column purification with the Qiagen DNA easy kit according to manufacturer's protocol (Qiagen, Valencia, CA), and purified DNA subjected to either PET-PCR or sen-qPCR as denoted above. To translate from real-time PCR signal to estimated parasite density, appropriate standard curves were prepared specific for each assay as described previously [20].

 Concentrations of HRP2, pan-*Plasmodium* lactate dehydrogenase (pLDH), and pan-*Plasmodium* aldolase (pAldolase) were calculated for each sample using the multiplexed antigen bead-based assay and extrapolation from assay signal to antigen concentration performed

using methods previously described [9]. For all laboratory data collected for analyses, it was assumed there was no sample contamination.

## Data analysis and malaria infection status classification

Principal components analysis (PCA) was performed using antigen concentration and log concentration for PCR presence/absence and infection level parasitemia (parasites/μL) based on five categories (none- 0, lowest—> 0–20, low—> 20–200, mid- > 200–2,000, high- > 2,000) using the prcomp function in R (R Foundation for Statistical Computing). Categories were selected on a log10 scale with the 200 p/μL as the benchmark, being the minimum parasite density RDT product testing employs [4]. As the Angola (microscopy dataset) were nearly all high density infections, Infection level categories for that dataset were the following: lowest = > 0–5,000; low = > 5,000–10,000; mid = >10,000–15,000; high = > 15,000. Conditional inference trees were constructed using the ctree function in the party package in R. Conditional inference trees were selected as a non-parametric regression analysis, which uses unbiased recursive partitioning on continuous, multivariate data to identify the most informative features (e.g., antigenic concentration) and quantitative thresholds for prediction. The decision trees were trained using leave-one-out cross validation using antigen concentration and log concentration as features to classify PCR presence/absence and infection level status as described above. Accuracy, sensitivity, and specificity of conditional inference trees was calculated using the following: accuracy = (true positive (tp) + true negative (tn)) / (tp + false positive (fp) + tn + false negative (fn)); sensitivity = tp / (tp + fn); specificity = tn / (fp + tn). All figures were created using the ggplot2 and cowplot libraries, unless otherwise stated. All R code for data analysis and figure generation is available at https://github.com/SESchmedes/plasmodium_falciparum_infection_status.

## Results

The *P. falciparum* prevalence of the study population was 28% from the 2016 Angola survey [17], and 4.0% from the Haiti study [18]. The mean age of participants for the 2015 Angola TES was 2.8 years with a median 3 years and range 7 months– 12 years. The mean age of participants for the 2016 Angola HF survey was 21 years with a median 15 years and range 1 month– 90 years. The mean age of participants for the 2012–2013 Haiti HF study was 17.4 years with a median 11 years and range 0–99 years.

HRP2, pLDH, and pAldolase concentrations were generated from a total of 826 dried blood spot samples collected in *P. falciparum* high-transmission (Angola) or low-transmission (Haiti) areas. Persons from the Angola TES had parasitemia determined by microscopy with a range of 2,175 to 184,464 parasites/μL (mean 49,230 parasites/μL); parasite densities for the Angolan health facility samples ranged from 0 to 43,290 parasites/μL (mean 618 parasites/μL); parasite densities for Haitian samples ranged from 0 to 18,463 parasites/μL (mean 908 parasites/μL) (Fig 1).

Conditional inference trees were trained using leave-one-out cross validation with HRP2, pAldolase, and pLDH concentrations (and log-transformed concentrations) for classification of PCR presence/absence status (Fig 2) and infection level (Fig 3, S1 Fig). Both HRP2 and pAldolase informed the models for PCR presence or absence for the full Angolan and Haitian datasets, but pLDH concentrations did not. Using the non-transformed antigen concentrations, the Angolan sen-qPCR data from the health facility survey only predicted one node (bifurcation point) at a pAldolase concentration greater than 325.3 pg/mL to predict *P. falciparum* presence or absence (Fig 2A), and the Haitian PET-PCR provided two nodes with the first at a HRP2 concentration of 183 pg/mL and the second at pAldolase at 274 pg/mL (Fig 2C).

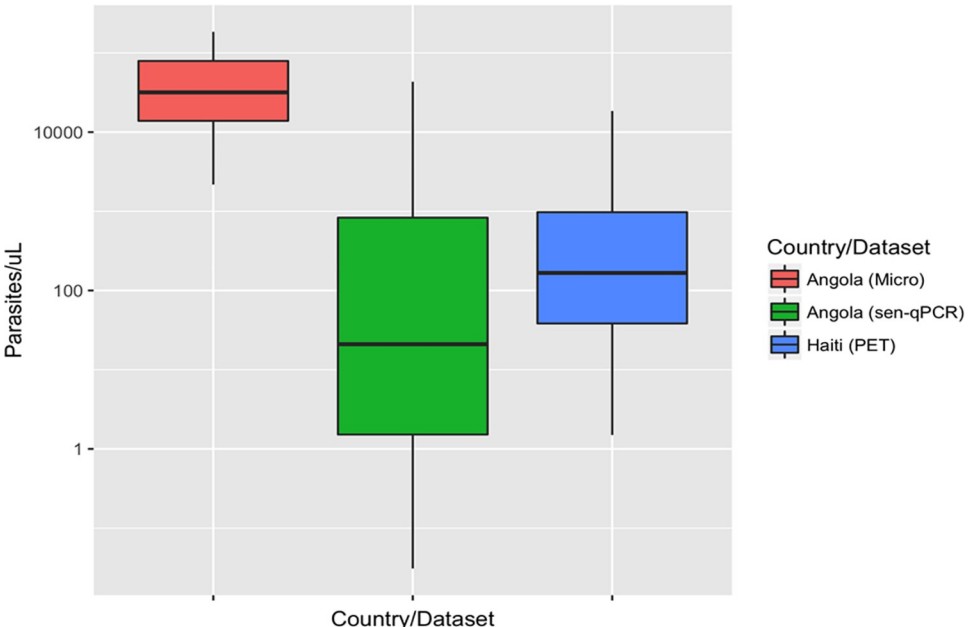

**Fig 1. Parasitemia distribution for each countries' dataset.** For each study, parasite density depicted as parasites/μL blood. Middle bar = median. Upper box hinge = 75[th] percentile. Lower box hinge = 25[th] percentile. Upper whisker = largest value no further than 1.5 * IQR (inter-quartile range or distance from first and third quartiles) from the hinge. Lower whisker = smallest value at most 1.5 * IQR from the hinge.

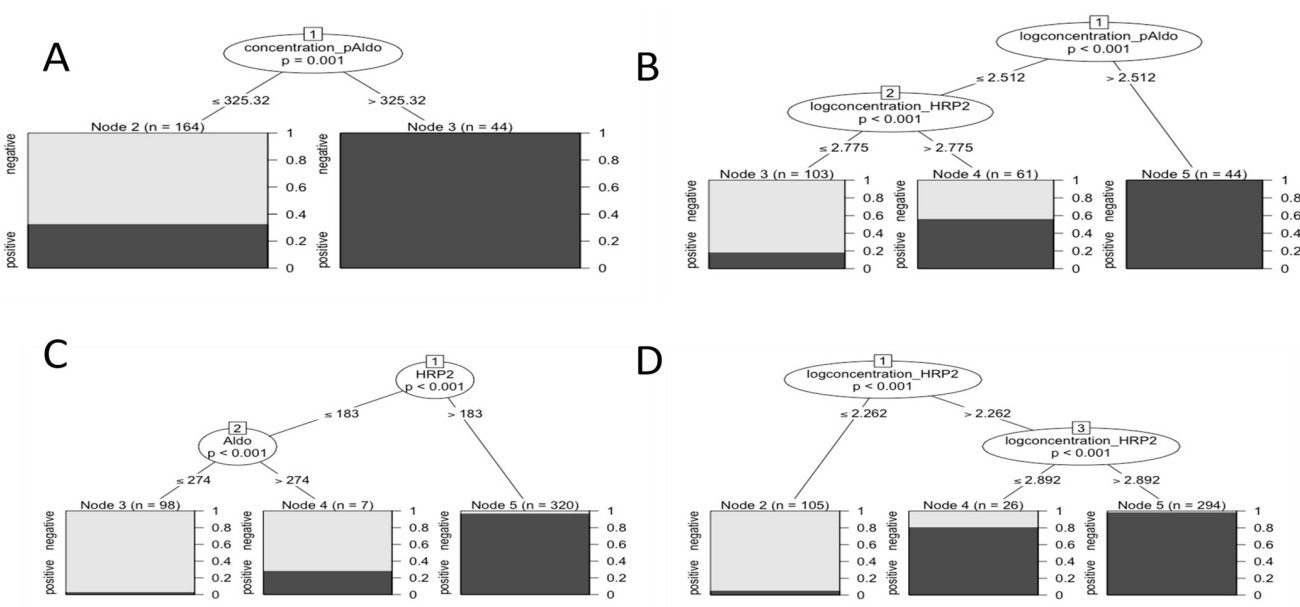

**Fig 2. Conditional inference trees using HRP2, pLDH, and pAldolase antigen concentration for classification of *Plasmodium falciparum* presence or absence as determined by PCR assay.** A) Angola (sen-qPCR). B) Angola (sen-qPCR), log scale. C) Haiti (PET-PCR). D) Haiti (PET), log scale. Y-axes at base of trees indicate probability of correct classification on a scale of 0.0 to 1.0.

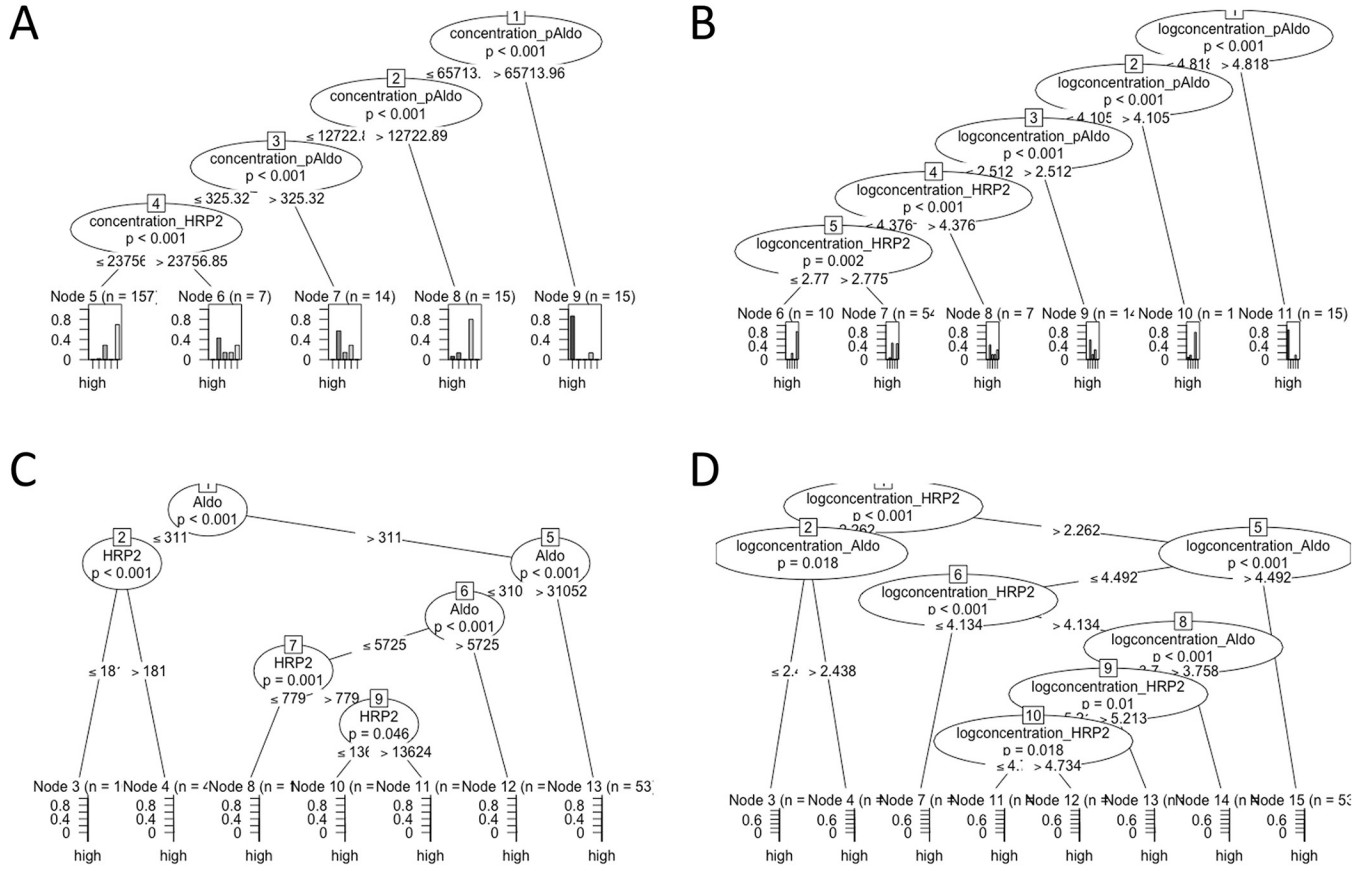

**Fig 3. Conditional inference trees using HRP2, pLDH, and pAldolase antigen concentration and log concentration for malaria infection level classification.** Infection level categories: None = 0 parasites/μL; Very low = > 0–20; Low = > 20–200; Mid = > 200–2,000; High = > 2,000. A) Angola (senqPCR). B) Angola (sen-qPCR), log. C) Haiti (PET-PCR). D) Haiti (PET), log. Y-axes at base of trees indicate probability of correct classification on a scale of 0.0 to 1.0.

When log-transforming the antigen data, additional prediction nodes were generated with the Angolan sen-qPCR predicting two nodes with the first of pAldolase at 325.1 pg/mL and second of HRP2 concentration at 595.7 pg/mL (Fig 2B). The log-transformed Haitian antigen data also provided two nodes for infection presence/absence, the first at HRP2 concentration of 182.8 pg/mL, and the second at a higher HRP2 concentration of 779.8 pg/mL (Fig 2D). PCR infection status was predicted with accuracies ranging from 73–96%, while infection level was predicted with accuracies ranging from 59–66% (Table 1).

**Table 1. Percent accuracies for malaria infection status prediction.**

| Country/Dataset | Attribute | Presence/Absence | Infection Level % accurate |
|---|---|---|---|
| | | % accurate (Se, Sp)* | |
| | Concentration | NA | 70 |
| Angola (microscopy) | Log concentration | NA | 72 |
| Angola (sen-qPCR) | Concentration | 73 (94%, 67%) | 66 |
| | Log concentration | 75 (73%, 78%) | 66 |
| Haiti (PET-PCR) | Concentration | 96 (97%, 95%) | 59 |
| | Log concentration | 96 (97%, 95%) | 66 |

* Se: sensitivity; Sp: specificity; Accuracy, Se, and Sp are based on correct classification of malaria parasite presence/absence by utilizing PCR result as gold standard

*P. falciparum* infection data was further modeled by conditional inference trees after sub-dividing into five categories based on levels of estimated parasite densities. As the Angola TES only enrolled participants based on a microscopically-confirmed parasite density above 2,000 p/μL, those data were not able to be evaluated using the same categorization scheme as the Angola (sen-qPCR) and Haiti (PET-PCR) datasets; therefore, higher concentration infection levels were used for resolution of level of infection (S1 Fig). In assessing the non-transformed antigen data, the Angolan sen-qPCR generated four nodes for infection level, with the first three from pAldolase and the fourth node from HRP2 (Fig 3A). The Haitian dataset provided further resolution with a primary node at 311 pg/mL of pAldolase, and further downstream nodes based on HRP2 or pAldolase concentrations (Fig 3C). As was the case for infection presence/absence, log-transformed antigen data provided more nodes for level of infection. For Angolan sen-qPCR data, log-transformed antigen data provided five nodes with the first three based on pAldolase concentration and the final two on HRP2 concentration (Fig 3B). Notably, log-transforming the Haitian antigen data now provided the first node at HRP2 (concentration of 182.8 pg/mL) with downstream nodes involving both HRP2 and pAldolase (Fig 3D). Percent accuracy for predicting *P. falciparum* infection level ranged between 59 and 72%, all which were lower than the accuracies of predicting simple presence/absence (Table 1).

For both presence/absence and infection level analyses with PCR data, pLDH did not provide significant decision nodes in the full dataset. However, if pAldolase data was removed, leaving only HRP2 and pLDH, then pLDH did provide nodes. An example is shown for the Angolan sen-qPCR dataset with pAldolase removed where pLDH provides the first nodes for both the non-transformed and transformed antigen concentrations (S2 Fig). The corresponding classification accuracy was approximately the same for both the non-transformed (75%) and log-transformed (75%) antigen data when compared to the full dataset with all three antigens included (Table 1). For all three studies, the pLDH and pAldolase concentrations were shown to correspond with each other (S3 Fig).

When performing principal component analysis (PCA) on the datasets, scatterplots displaying the concordance between the first and second principal components (PC) showed a degree of clustering based on infection presence/absence as well as level of infection. For data from PCR assays, no discernable clustering was observed with non-transformed antigen data for infection presence/absence (S4A and S4C Fig), but when antigen data were log-transformed, the Haiti PET-PCR infection presence was strongly connected to lower values of PC1 (which explained 81.1% of variance) (S4D Fig). When assessing data by level of infection, the non-transformed antigen data again did not show defined visual clustering (Fig 4A and 4C). Higher PC1 values were strongly connected to higher parasite densities for the log-transformed antigen data from Angola (Fig 4B), but the inverse was true for the Haiti dataset with lower PC1 values connected with the 'mid' and 'high' infection levels. Infection level categories by microscopy did not show any visual clustering by scatterplots of PC1 and PC2 (S5 Fig).

## Discussion

Our results suggest that machine learning algorithms can be trained using quantitative malaria antigen data to reliably predict *P. falciparum* presence/absence and as well as different levels of peripheral parasite densities. Antigen detection is utilized globally for diagnosis of malaria by RDTs, but these are designed to detect clinically-relevant parasite densities, and only provide a binary result [3, 4]. Additionally, standard RDT use would also require a point-of-contact action (i.e. administering anti-malarial drugs) upon a positive result, and the multiplex antigen data and analyses presented here would be more utilized for epidemiological purposes. By being able to categorize sample sets into levels of estimated parasite densities based on

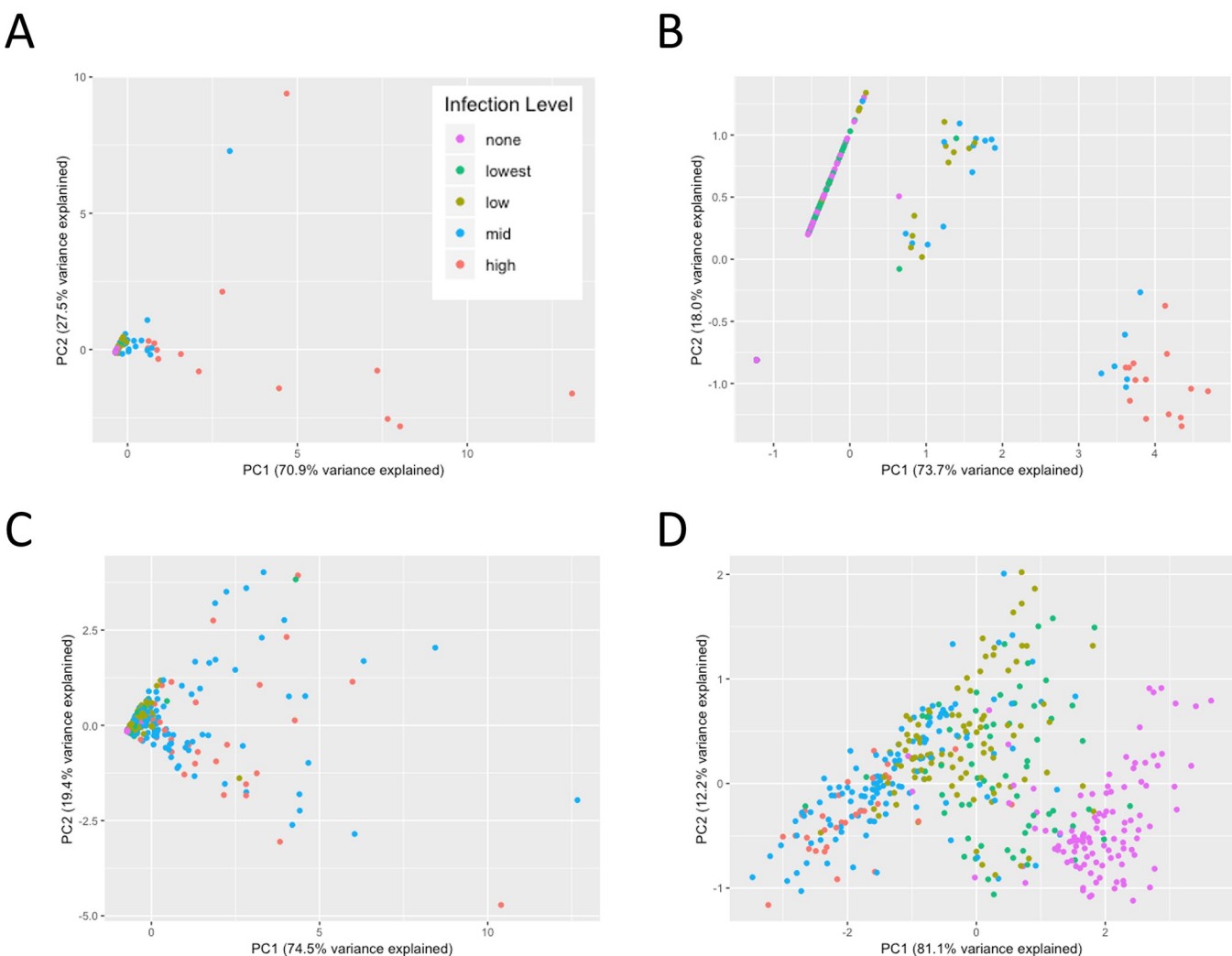

**Fig 4. Principal components analysis of HRP2, pLDH, and pAldolase concentrations and log concentrations for infection level using qPCR.** Infection level categories: None = 0 parasites/μL; Very low = > 0–20; Low = > 20–200; Mid = > 200–2,000; High = > 2,000. A) Angola (sen-qPCR). B) Angola (sen-qPCR), log scale. C) Haiti (PET-PCR). D) Haiti (PET-PCR), log scale.

multiplex antigen data alone, an additional benefit could arise by being able to select samples within these higher levels for greater success with DNA-based assays. The ability to use quantitative antigen concentrations to train machine learning algorithms to predict peripheral parasite densities represents a novel step forward for these efforts.

Collection of quantitative multiplex antigen data presents many advantages to other laboratory assays for the estimation of malaria status from a patient sample. These immunoassays are formatted to a 96-well format, the per-sample cost is approximately an order of magnitude less than nucleic-acid based assays, and hands-on time in the laboratory is short due to the simplicity of the antigen detection assays [9, 10]. For the datasets available for this study, quantitative data for three *Plasmodium* antigens was available: pan-*Plasmodium* LDH and aldolase antigens (pLDH and pAldolase, respectively), and *P. falciparum*-specific HRP2. As all three datasets were specifically capturing *P. falciparum* infections by microscopy or PCR assays, this panel of three antigens was appropriate for this investigation. However, non-falciparum infections have been reported in both Angola [9, 21] and Haiti [22], so the possibility also exists that

non-detected mixed infections with *P. vivax*, *P. malariae*, or *P. ovale* (only in Angola) could have skewed the pan-*Plasmodium* antigen concentrations beyond what would be expected for a *P. falciparum*-only infection. However, as a proportion of the total malaria burden on these populations, non-falciparum infections are rare in these two countries, so presence of mixed infections would likely not have influenced the models. With quantitative data potentially available for other malaria antigen targets not utilized here, these similar machine learning approaches could be expanded to be even more robust in predicting *P. falciparum* infection or be modeled against infection with another of the human malarias.

In assessing the model input of the pan-*Plasmodium* antigens for the complete datasets, pLDH was only informative for the Angola TES study (infection detected by microscopy), whereas pAldolase was informative for nearly every other decision tree with both non-transformed and log-transformed antigen concentrations. It may not be surprising that one of these pan- antigens would "out-compete" the other as high collinearity was observed in absolute concentrations between these two targets in the same sample (S3 Fig) [9]. As an example, when the pAldolase data was removed from the Angolan health facility dataset, the pLDH replaced pAldolase as the first node on the decision tree. As both are metabolic enzymes in *Plasmodia spp*., though concentrations of these two antigens would be expected to be largely concordant, they both may provide unique information for different strains of *P. falciparum* parasites which may express slightly different isoforms of these two antigens [23, 24]. By far, the most informative input for the models creating the most nodes was the HRP2 antigen, which is abundantly expressed during blood-stage *P. falciparum* infection [25]. This was true in training models for both *P. falciparum* presence/absence as well as level of parasite density during infection. The added advantage of being able to measure signals for multiple antigens at a time is consistent with previous reports showing that HRP2/LDH ratios are predictive of determining active from recently cleared infection [26].

When compared to models for parasite presence/absence, modeling for discrete infection levels had noticeably lower accuracy. This was not surprising as blood-stage malaria infection is characteristic of billions of parasites and high amounts of antigen being produced, so an identified infection typically has very high amounts of these antigens in the host without precise gradations. Additionally, the "very low" (1–20 parasites/μL) and "low" (20–200 parasites/μL) categories of *P. falciparum* infection are both under the parasite density levels evaluated by the World Health Organization RDT evaluation program for product qualification [4]. The highest accuracy, up to 96%, was observed for prediction of PCR presence/absence in Haiti, and this could be explained by the low-transmission setting in this country; individuals would have been less likely to have had a recent infection with antigen concentrations creating "noise" that makes it more difficult to distinguish from active infection.

A limitation to this study was that datasets from each of the three studies provided different sample sizes in terms of number of persons infected with *P. falciparum* and utilized different enrollment criteria and samples from persons with different exposure histories. Additionally, the only sample type utilized in these surveys was DBS, and during drying and storage, potential degradation of antigen or DNA may have occurred to understate the quantity of these biomarkers. Different PCR assays were used for Haiti and Angola, and while both estimated parasite densities from quantity of DNA in the samples, comparison of classification trees between these two sample sets should consider the differences in PCR assays. A more recently detected phenomenon of *P. falciparum* strains with deletions or alterations of the *pfhrp2* gene has been seen in numerous countries but was not evaluated in this study [8]. However, these deletions have not been reported in Haiti [27], and only reported at very low levels in Angola [9], so these potential deletions likely did not affect our analyses. High-transmission areas (like Angola) might not perform as well using this model compared to low-transmission areas (like

Haiti) due to lingering HRP2 antigen in circulation [5, 7], which could negatively impact specificity estimates.

Future studies on larger datasets should address optimal statistical tests and machine learning models for infection status prediction, as well as employ methods to correct for dataset imbalance. Conditional inference trees were selected for use in this study to perform a non-parametric regression analysis as a method for unbiased recursive partitioning to easily identify the most informative antigen for the model, in addition to predictive quantitative thresholds of antigenic concentrations. As such, it could not be stated that this approach utilized here would produce optimal accuracy, and further investigation of other statistical approaches, such as k-nearest neighbor regression, linear discrimination analysis, random forest, gradient boosting, or finite mixture models, should be conducted on future antigenic concentration datasets collected addressing the limitations stated above.

This study provides a pilot methodology and the results can be used to design and conduct additional studies. Specifically, future validation studies should have datasets with: substantial numbers of negative and positive samples with a wide range of parasite densities; molecular detection/parasite density measured using a variety of quantitative PCR techniques; and samples from different geographical areas (including pre-elimination, low-transmission and high-transmission countries/regions). Further investigation of machine learning approaches could provide greater resolution for determination of infection status from quantitative antigen data to support malaria surveillance activities and epidemiologic studies.

## Supporting information

**S1 Fig. Conditional inference trees using HRP2, pLDH, and pAldolase antigen concentration for malaria infection level classification by microscopy.** Infection level categories: Lowest = > 0–5,000; Low = > 5,000–10,000; Mid = >10,000–15,000; High = > 15,000. A) Angola (microscopy). B) Angola (microscopy), log scale. Y-axes at for all plots indicate probability of correct classification on a scale of 0.0 to 1.0.
(TIF)

**S2 Fig. Conditional inference trees using only HRP2 and pLDH data with pAldolase removed for classification of *Plasmodium falciparum* presence or absence as determined by PCR assay.** Data shown for Angola (sen-qPCR) classification with antigen concentrations on non-transformed scale on left and log-transformed on right. Y-axes at base of trees indicate probability of correct classification on a scale of 0.0 to 1.0.
(TIF)

**S3 Fig. Comparison of concentrations of pLDH and pAldolase antigens from persons enrolled among the three surveys.** Antigen concentration data shown for the Angola 2015 TES (A), Angola 2016 health facility (B), and Haiti bednet (C) studies with concentration of pAldolase on x-axis and pLDH on y-axis for each.
(TIF)

**S4 Fig. Principal components analysis of HRP2, pLDH, and pAldolase concentrations and log concentrations for qPCR presence/absence.** A) Angola (sen-qPCR). B) Angola (sen-qPCR), log scale. C) Haiti (PET-PCR). D) Haiti (PET), log scale.
(TIF)

**S5 Fig. Principal components analysis of HRP2, pLDH, and pAldolase concentrations for infection level using microscopy.** Infection level categories: Lowest = > 0–5,000; Low = > 5,000–10,000; Mid = >10,000–15,000; High = > 15,000. A) Angola (microscopy). B) Angola

(microscopy), log scale.
(TIF)

**S1 File. Inclusivity in global research.**
(DOCX)

## Acknowledgments

**Disclaimer:** The findings and conclusions in this article are those of the authors and do not necessarily represent the views of the Centers for Disease Control and Prevention or the US Agency for International Development.

## Author Contributions

**Conceptualization:** Sarah E. Schmedes, Mateusz Plucinski, Eric Rogier.

**Data curation:** Sarah E. Schmedes, Eric Rogier.

**Formal analysis:** Sarah E. Schmedes, Mateusz Plucinski.

**Project administration:** Rafael P. Dimbu, Laura Steinhardt, Jean F. Lemoine, Michelle A. Chang, Mateusz Plucinski.

**Writing – original draft:** Sarah E. Schmedes, Eric Rogier.

**Writing – review & editing:** Mateusz Plucinski.

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
