## [Decision Letter · Decision Letter 0]

19 Dec 2021

PONE-D-21-34503Predicting Plasmodium falciparum infection status in blood using a multiplexed bead-based antigen detection assay and machine learning approachesPLOS ONE

Dear Prof. Rogier,

Thank you for submitting your manuscript to PLOS ONE. After careful consideration, we feel that it has merit but does not fully meet PLOS ONE’s publication criteria as it currently stands. Therefore, we invite you to submit a revised version of the manuscript that addresses the points raised during the review process.

A number of points have been raised by the reviewers which a revised manuscript would need to address. These include:The detail on the statistical methods (how do these analysis work) and descriptions of related specific terms are lacking. Please provide clear description of these so that the non expert can follow clearly.Reviewer 2 raises a number of relevant questions on why this particular set of statistical analysis were used as compared to others and the origin of these tests. Please consider whether the title is accurate and appropriate. If you believe it is, please provide justification.Please discuss and justify why this particular set of analysis was chosen to be done over other options. Please discuss the advantages and limitations over other statistical analysis techniques.ROC curves should be provided.Please consider adding finite fixture models. This is an opportunity to strengthen the manuscript.Information on the study samples is limited. Please provide additional information on sample parameters and also discuss this in light of the study findings as suggested by the reviewers.Please address the use of the Angloa dataset in the absence of infection data and either justify or remove the dataset.Please address the remaining minor comments and take on board suggestions for additional discussion on the studies findings. This includes the impact of low parasite density samples on the studies findings.

We look forward to receiving your revised manuscript.

Kind regards,

Danny W Wilson

Academic Editor

PLOS ONE

Journal Requirements:

2. Please include a complete copy of PLOS’ questionnaire on inclusivity in global research in your revised manuscript. Our policy for research in this area aims to improve transparency in the reporting of research performed outside of researchers’ own country or community. The policy applies to researchers who have travelled to a different country to conduct research, research with Indigenous populations or their lands, and research on cultural artefacts. The questionnaire can also be requested at the journal’s discretion for any other submissions, even if these conditions are not met.  Please find more information on the policy and a link to download a blank copy of the questionnaire here: https://journals.plos.org/plosone/s/best-practices-in-research-reporting. Please upload a completed version of your questionnaire as Supporting Information when you resubmit your manuscript

4. Please provide additional details regarding participant consent. In the ethics statement in the Methods and online submission information, please ensure that you have specified (1) whether consent was informed and (2) what type you obtained (for instance, written or verbal, and if verbal, how it was documented and witnessed). If your study included minors, state whether you obtained consent from parents or guardians. If the need for consent was waived by the ethics committee, please include this information.

The authors acknowledge partial support from the Bioinformatics Fellowship Program administered by the Association of Public Health Laboratories (APHL) and funded by the CDC. S.S. is supported in part by the Bioinformatics Fellowship Program administered by the APHL and funded by the CDC.

MP was supported by the U.S. President’s Malaria Initiative

6. We note that you have indicated that data from this study are available upon request. PLOS only allows data to be available upon request if there are legal or ethical restrictions on sharing data publicly. For more information on unacceptable data access restrictions, please see http://journals.plos.org/plosone/s/data-availability#loc-unacceptable-data-access-restrictions. 

Additional Editor Comments:

Figures were quite blurred. Please review the PLOS One figure requirements and see whether Figures can be uploaded in a higher resolution format.Do you intend to provide the code and data that underline this study?

Reviewers' comments:

Reviewer's Responses to Questions

**Comments to the Author**

1. Is the manuscript technically sound, and do the data support the conclusions?

Reviewer #1: Yes

Reviewer #2: Partly

2. Has the statistical analysis been performed appropriately and rigorously? 

Reviewer #1: I Don't Know

Reviewer #2: No

3. Have the authors made all data underlying the findings in their manuscript fully available?

Reviewer #1: Yes

Reviewer #2: No

4. Is the manuscript presented in an intelligible fashion and written in standard English?

Reviewer #1: Yes

Reviewer #2: Yes

5. Review Comments to the Author

Reviewer #1: This study examines whether the quantitative measurements of Plasmodium antigen in human exposed blood samples can be used to train machine learning algorithms to categorize patients by parasite presence or absence and parasite density level in order to achieve greater nuance in malaria diagnostics using antigen based assays rather than more costly DNA amplification methodologies. Blood samples from different cohort comprising symptomatic children and febrile or non-febrile adults were used to determine parasitemia by different quantitative methods, and antigen levels for three different antigens were determine by a bead-based quantitative assay. Conditional inference trees and PCA were performed for classification into presence or absence of parasites and into pre-defined parasite levels based on transformed and un-transformed antigen levels. Prediction of PCR infection status and parasitemia levels varied depending on antigen transformation of data.

This pilot study provides encouraging results to show that machine learning algorithms can be trained using quantitative antigen data in order to predict infection status and parasitemia levels. Results and limitations of the study and study cohort were appropriately discussed.

Minor points:

Given that the machine based learning algorithms are trained by continuous antigen data derived from dried blood spots, could the authors comment or speculate on how this would impact on the assessment of fresh blood samples in POC health care settings (i.e. would antigen measurement and DNA extraction from dried blood spots potentially underestimate parasite density due to poor antigen and DNA extraction?).

Results, lines 146ff describe the mean number of parasites/µl whereas figure 1 shows median. Could the authors make this consistent, i.e. quote median in text or show mean in figure.

Table 1: Can the authors please define sensitivity and specificity in the table notes

Font size in all figures and tables should be increased

Figure S3: Is this a Spearman or Pearson’s correlation

Reviewer #2: The paper entitled “Predicting Plasmodium falciparum infection status in blood using a multiplexed bead-based antigen detection assay and machine learning approaches” is an application of conditional inference trees to malaria infection data. The paper is interesting and, as far as I know, it is the first time that this type of methodology was applied to malaria data; other machine learning techniques such as random forests were used in searching protective immunity against clinical malaria (doi: 10.1371/journal.pcbi.1005812) and this should be acknowledged in the Introduction. However, I think an extensive revision of the paper is needed in order to make the take-home message more compelling and crystal clear. Below please find my specific comments:

- The title does not reflect the content of the paper. First, infection levels were also predicted in the study. Second, it was only applied a single method (i.e., conditional inference trees) to predict infection status and levels and hence the use of the plural ''approaches'' is not appropriate. Third, if one tracks down the theoretical developments of conditional inference trees, they were published in statistical rather machine learning journals. Therefore, conditional inference trees are more statistical learning techniques than machine learning ones.

- The motivation of this study should be rephrased more clearly. Is the motivation on the use of the methodology or the use of data from multiplex bead assays? I guess it is the methodology but then why to use conditional inference trees and exclude other existing methodologies to tackle the same classification problem?

- In the introduction, it is important to be clear that absent production of the antigen target (line 71) is mostly for the HRP2 case. As far as I know, there are no reports of gene deletions for Aldolase and LDH. This might be obvious for most malaria researchers but for less specialized audience, I would write that LDH, Aldolase, and HRP2 are the proteins used in current pan-malaria and pf-malaria RDTs.

- Please provide data about the prevalence of infection for the Haiti study and the second study from Angola. It would be useful to provide information (mean, median, range, etc) about the age of the participants from each study. This increases the interpretability of the results.

- In Figure 1 (Angola sen qPCR), there are 25% of the infections that have parasitemia below 1 mu/l. Can we trust these low levels of parasite density? What is the lower level of detection above which one can trust the respective parasitemia quantification? This point is important to clarify given that the limited performance of the conditional inference trees might be caused by these infections with low parasitemia.

- Is there any rationale to divide infection levels into the 5 categories used? If this categorization is completely arbitrary, this should be clearly stated. Otherwise, provide a rationale (maybe related to the expected sensitivity to RDT as function of parasitemia).

- In the materials & methods, provide a brief explanation about conditional inference trees, how they are constructed and interpreted. This increases readability of the paper to a less specialized audience.

- It is worth mentioning that conditional inference trees are dependent on the scale of covariates/features. This is a limitation of the methodology that should be acknowledged. This limitation could have been avoided by using other methodologies, such as random forest or XGBoost, which are invariant to change of scale. Why were not these methodologies applied to the same data?

- It is also unclear whether simpler and more common approaches such as logistic regression, probit regression or other generalized models for binary/categorical data could perform equally well in the same data. Linear discrimination analysis is also another population alternative for classification problems using multivariate data.

- It was used a leave-one-out cross-validation procedure. This allows the estimation of the sensitivity (Se) and specificity (Sp) shown in Table 1. But I think 5-fold or 10-fold cross-validation provides a better idea of how robust (or uncertain) accuracy, Sp and Se estimates are. Please define accuracy, Sp and Se for a less specialized audience.

- To complement the presented accuracy measures of the model predictions, the ROC curves should be also presented (in the main text) and the respective area under the curve calculated.

- I like the idea of having cutoffs in the covariates/features. This reminds what malaria epidemiologist do in serological data analysis where a cutoff is used to define seronegative and seropositive population. From a perspective of responsible and explicable machine learning, I recommend to fit finite mixture models (Gaussian or non-Gaussian) to the antigen data, check whether there are multiple latent populations (e.g, antigen-negative plus multiple antigen-positive levels), and whether the cutoffs derived from conditional inference trees are related to the discrimination between these latent populations. Particularly flexible finite mixture models are the ones based on the Skew-normal and Skew-t distributions as described in Domingues et al (doi: 10.1101/2021.03.08.21252807). This additional analysis takes the paper into a whole new level.

- In Table 1, accuracy for the infection level data should be discriminated per infection level. I bet misclassification comes mostly from categories related to low parasitemia infections.

- In Table 1, it is interesting that Sp seems to be lower in Angola than in Haiti. I bet this is related to a higher transmission in Angola than in Haiti. This is an interesting finding that deserves exploration and discussion.

- In the text, it says the accuracy for the infection levels ranged from 59% and 72%, but the estimates in Table 1 do not show any estimate equal to 72%. Hopefully, this is just a typo.

- I am confused that “As the Angola TES 185 only enrolled participants based on a microscopically-confirmed parasite density above 2,000 p/�L, those data were not able to be evaluated in this categorization scheme” (lines 184—186). It seems this dataset was not used at all for prediction given that it could not also be used for infection status prediction . If that is the case, the paper needs to be totally revised to remove any reference to this dataset (including Figures and Supplementary Figures).

- What was the rationale to include a principal component analysis (PCA) as it is not use to predict infection status and levels? Given that the objective is related to a classification problem, why not to use a related multivariate technique such as linear discriminant analysis as suggested above?

- With respect to PCA, I cannot observe that higher values of PC1 reflect high infection levels for Figure 4D (lines 224-225).

- Figures: unfortunately, I could not make a better assessment of the figures due to their low resolution. However, I think violin plots or related plots provide a more informative way of visualizing the data instead of the boxplots shown in Figure 1. In the same Figure, it should be clearly stated in the figure legend that non-infected individuals are not represented in the plots. The remaining Figures are unreadable. What is it plotted in the y axis of the plots at the bottom of the trees? All figure legends should be expanded to be more informative.

- Code and data sharing: to increase replication of the study by other researchers, authors should consider to share their data and code with the community.

6. PLOS authors have the option to publish the peer review history of their article (what does this mean?). If published, this will include your full peer review and any attached files.

Reviewer #1: No

Reviewer #2: No

---

## [Author Response · Author response to Decision Letter 0]

1 Jul 2022

Response to Editor and Reviewers

Additional Editor Comments:

• Figures were quite blurred. Please review the PLOS One figure requirements and see whether Figures can be uploaded in a higher resolution format.

- This has been done.

• Do you intend to provide the code and data that underline this study?

- Yes, we have now provided the code on Github, and indicated this at the end of Methods. We have also added a statement to the end of Methods that “Data used in these analyses is available upon request to the corresponding author”. 

Reviewer #1: This study examines whether the quantitative measurements of Plasmodium antigen in human exposed blood samples can be used to train machine learning algorithms to categorize patients by parasite presence or absence and parasite density level in order to achieve greater nuance in malaria diagnostics using antigen based assays rather than more costly DNA amplification methodologies. Blood samples from different cohort comprising symptomatic children and febrile or non-febrile adults were used to determine parasitemia by different quantitative methods, and antigen levels for three different antigens were determine by a bead-based quantitative assay. Conditional inference trees and PCA were performed for classification into presence or absence of parasites and into pre-defined parasite levels based on transformed and un-transformed antigen levels. Prediction of PCR infection status and parasitemia levels varied depending on antigen transformation of data.

This pilot study provides encouraging results to show that machine learning algorithms can be trained using quantitative antigen data in order to predict infection status and parasitemia levels. Results and limitations of the study and study cohort were appropriately discussed.

Minor points:

Given that the machine based learning algorithms are trained by continuous antigen data derived from dried blood spots, could the authors comment or speculate on how this would impact on the assessment of fresh blood samples in POC health care settings (i.e. would antigen measurement and DNA extraction from dried blood spots potentially underestimate parasite density due to poor antigen and DNA extraction?).

- The reviewer brings up an important point here regarding protein and DNA integrity from different sample types. Additional text has been added to the limitations section of Discussion (Line 297) to point out that only DBS were used in these studies, and the potential for antigen/DNA degradation: “ Additionally, the only sample type utilized in these surveys was DBS, and during drying and storage, potential degradation of antigen or DNA may have occurred to understate the quantity of these biomarkers.”

Results, lines 146ff describe the mean number of parasites/µl whereas figure 1 shows median. Could the authors make this consistent, i.e. quote median in text or show mean in figure.

- We report the mean in the text because it is more informative than the median (for example, some of the medians are 0). Please note that the y-axis in the Figure is on a log-scale.

Table 1: Can the authors please define sensitivity and specificity in the table notes

- Additional text has been added to the Table 1 footnote to explain this further: “Accuracy, Se, and Sp are based on correct classification of malaria parasite presence/absence by utilizing PCR result as gold standard”.

Font size in all figures and tables should be increased

 - This has been done 

Figure S3: Is this a Spearman or Pearson’s correlation

- The use of the word ‘correlation’ is not the most appropriate here, as we do not provide statistical tests to assess true correlation and significance. This word has been replaced in both the text and figure legend. 

Reviewer #2: The paper entitled “Predicting Plasmodium falciparum infection status in blood using a multiplexed bead-based antigen detection assay and machine learning approaches” is an application of conditional inference trees to malaria infection data. The paper is interesting and, as far as I know, it is the first time that this type of methodology was applied to malaria data; other machine learning techniques such as random forests were used in searching protective immunity against clinical malaria (doi: 10.1371/journal.pcbi.1005812) and this should be acknowledged in the Introduction. 

- We thank the reviewer for this suggestion, and this study has been acknowledged in the Introduction and is now reference #15. 

However, I think an extensive revision of the paper is needed in order to make the take-home message more compelling and crystal clear. Below please find my specific comments:

- The title does not reflect the content of the paper. First, infection levels were also predicted in the study. Second, it was only applied a single method (i.e., conditional inference trees) to predict infection status and levels and hence the use of the plural ''approaches'' is not appropriate. Third, if one tracks down the theoretical developments of conditional inference trees, they were published in statistical rather machine learning journals. Therefore, conditional inference trees are more statistical learning techniques than machine learning ones.

 - The authors thank the reviewers for the insight. We agree with the reviewer that the main narrative of the paper is not to assess the ideal machine learning approach to this sort of dataset. Our objective was to show that a machine learning approach (like conditional inference trees) can be effective at classifying samples by infection status. There is a large overlap between “statistical approach” and “machine learning approach”, and both could apply here. From our perspective, “infection status” is inclusive of both positivity/negativity as well as infection level.

- The motivation of this study should be rephrased more clearly. Is the motivation on the use of the methodology or the use of data from multiplex bead assays? I guess it is the methodology but then why to use conditional inference trees and exclude other existing methodologies to tackle the same classification problem?

 - Thank you for pointing out this lack of clarity in our report. Our motivation was to apply an appropriate statistical tool (conditional inference trees) that was chosen in advance and apply it to a classification problem using data from multiplex bead assays for antigen detection. Our motivation was decidedly not to compare different methodologies and choose the most appropriate one – that would be well outside the scope of this work. However, we appreciate this comment and have added an additional paragraph near the end of Discussion explicitly stating the possibility of using alternative statistical and machine learning tools to address the same classification problem. We hope to explore other methods in future work.

- In the introduction, it is important to be clear that absent production of the antigen target (line 71) is mostly for the HRP2 case. As far as I know, there are no reports of gene deletions for Aldolase and LDH. This might be obvious for most malaria researchers but for less specialized audience, I would write that LDH, Aldolase, and HRP2 are the proteins used in current pan-malaria and pf-malaria RDTs.

 - The authors agree with this comment and that this should be better clarified. We add additional text here to accentuate that these three antigens are the current targets for malaria RDTs, and also clarify that only HRP2 is known to have gene deletions that would affect RDT results. 

- Please provide data about the prevalence of infection for the Haiti study and the second study from Angola. It would be useful to provide information (mean, median, range, etc) about the age of the participants from each study. This increases the interpretability of the results.

- The authors agree this is useful information for the readers’ interpretation. We have now added the requested information to the beginning of the Results section. 

- In Figure 1 (Angola sen qPCR), there are 25% of the infections that have parasitemia below 1 mu/l. Can we trust these low levels of parasite density? What is the lower level of detection above which one can trust the respective parasitemia quantification? This point is important to clarify given that the limited performance of the conditional inference trees might be caused by these infections with low parasitemia.

- To provide the reader with additional information, we have added the analytical sensitivity of this qPCR assay to line 119 of Methods with appropriate reference: “…, with an analytical sensitivity of 0.02 parasites/µL (19).”

- Is there any rationale to divide infection levels into the 5 categories used? If this categorization is completely arbitrary, this should be clearly stated. Otherwise, provide a rationale (maybe related to the expected sensitivity to RDT as function of parasitemia).

- The reviewer is correct regarding the RDT test reliability assumption, and the categories were selected on a log10 scale with the 200p/uL as the benchmark being the minimum parasite density RDT product testing employs. The authors have added additional text to line 137 in Methods to explain this. 

- In the materials & methods, provide a brief explanation about conditional inference trees, how they are constructed and interpreted. This increases readability of the paper to a less specialized audience.

- The authors have added an enhanced explanation of conditional inference trees in the “Data analysis and malaria infection status classification” section.

- It is worth mentioning that conditional inference trees are dependent on the scale of covariates/features. This is a limitation of the methodology that should be acknowledged. This limitation could have been avoided by using other methodologies, such as random forest or XGBoost, which are invariant to change of scale. Why were not these methodologies applied to the same data?

- The authors appreciate the comment and have added an additional paragraph to Discussion that proposes future alternate statistical approaches including k-nearest neighbor regression, linear discrimination analysis, random forest, gradient boosting, or finite mixture models.

- It is also unclear whether simpler and more common approaches such as logistic regression, probit regression or other generalized models for binary/categorical data could perform equally well in the same data. Linear discrimination analysis is also another population alternative for classification problems using multivariate data.

- The authors appreciate this comment, and will refer the reviewer to the comment above regarding other statistical approaches. The authors have updated Discussion to explicitly mention alternative statistical approaches. 

- It was used a leave-one-out cross-validation procedure. This allows the estimation of the sensitivity (Se) and specificity (Sp) shown in Table 1. But I think 5-fold or 10-fold cross-validation provides a better idea of how robust (or uncertain) accuracy, Sp and Se estimates are. Please define accuracy, Sp and Se for a less specialized audience.

- In this study, leave-one-out cross validation was used to maximize the training set since our datasets were relatively small. 10-fold validations will be used on future, larger datasets to minimize overfitting. Definitions of accuracy, Sp, and Se are now fully described in Methods section.

- To complement the presented accuracy measures of the model predictions, the ROC curves should be also presented (in the main text) and the respective area under the curve calculated.

- To our knowledge, it is not possible to calculate ROC curves for classification problems where there is more than one predictor variable. 

- I like the idea of having cutoffs in the covariates/features. This reminds what malaria epidemiologist do in serological data analysis where a cutoff is used to define seronegative and seropositive population. From a perspective of responsible and explicable machine learning, I recommend to fit finite mixture models (Gaussian or non-Gaussian) to the antigen data, check whether there are multiple latent populations (e.g, antigen-negative plus multiple antigen-positive levels), and whether the cutoffs derived from conditional inference trees are related to the discrimination between these latent populations. Particularly flexible finite mixture models are the ones based on the Skew-normal and Skew-t distributions as described in Domingues et al (doi: 10.1101/2021.03.08.21252807). This additional analysis takes the paper into a whole new level.

- We appreciate this suggestion from the reviewer and will refer to the added comment above which the authors have updated the Discussion section to explicitly mention alternative statistical approaches. Using and/or comparing alternative approaches is outside the scope of this work. Specifically, this current unit of work attempts to move away from more simple antigen positive/negative approaches and maintain antigen concentrations as continuous. 

- In Table 1, accuracy for the infection level data should be discriminated per infection level. I bet misclassification comes mostly from categories related to low parasitemia infections.

- We thank the reviewer for the suggestion here, but due to relatively smaller datasets utilized here, these accuracy estimates are unable to be reliably generated when sub-dividing the dataset by the 5 infection levels. We have added in Discussion the need to take these (and alternative) approaches on larger datasets. 

- In Table 1, it is interesting that Sp seems to be lower in Angola than in Haiti. I bet this is related to a higher transmission in Angola than in Haiti. This is an interesting finding that deserves exploration and discussion.

- The authors agree with the reviewer, and have revised the last sentence in the limitations paragraph to point this out directly: “High-transmission areas (like Angola) might not perform as well using this model compared to low-transmission areas (like Haiti) due to lingering HRP2 antigen in circulation (5, 7), which could negatively impact specificity estimates.” 

- In the text, it says the accuracy for the infection levels ranged from 59% and 72%, but the estimates in Table 1 do not show any estimate equal to 72%. Hopefully, this is just a typo.

- Yes, this was a typo, and has been corrected.

- I am confused that “As the Angola TES 185 only enrolled participants based on a microscopically-confirmed parasite density above 2,000 p/�L, those data were not able to be evaluated in this categorization scheme” (lines 184—186). It seems this dataset was not used at all for prediction given that it could not also be used for infection status prediction. If that is the case, the paper needs to be totally revised to remove any reference to this dataset (including Figures and Supplementary Figures).

- Because the Angola microscopy dataset only contained samples >2,000, that was automatically in our “high” infection level status compared to the other Angola (sen-PCR) and Haiti datasets. As the Angola (microscopy) dataset used a different category, those results were only include in the Supplemental. The authors have edited the figure legend to reflect the correct category levels and the text referred to in this comment. The authors have now included the accuracy for this dataset in the main Table 1.

- What was the rationale to include a principal component analysis (PCA) as it is not use to predict infection status and levels? Given that the objective is related to a classification problem, why not to use a related multivariate technique such as linear discriminant analysis as suggested above?

- Please see comment above – the authors have updated the discussion to explicitly mention alternative statistical approaches, including linear discriminant analysis. For this current study, PCA was utilized to visualize sub-populations of infection status (as determined by PCR data) as a factor of antigen concentrations. 

- With respect to PCA, I cannot observe that higher values of PC1 reflect high infection levels for Figure 4D (lines 224-225).

- The authors apologize for the poor resolution of the originally submitted figures, and we have uploaded new figures with higher resolution. The reviewer is correct in that lower PC1 values correspond to higher PCR-determined parasite densities in Figure 4D, and have revised Results text to reflect this.

- Figures: unfortunately, I could not make a better assessment of the figures due to their low resolution. However, I think violin plots or related plots provide a more informative way of visualizing the data instead of the boxplots shown in Figure 1. In the same Figure, it should be clearly stated in the figure legend that non-infected individuals are not represented in the plots. The remaining Figures are unreadable. What is it plotted in the y axis of the plots at the bottom of the trees? All figure legends should be expanded to be more informative.

- The authors apologize for the poor figure resolution, and have resubmitted with higher resolution figures. The authors appreciate the suggestion for violin plots, but prefer to stick with boxplots as they accurately depict the data and are more intuitive for the reader. The y-axis for the trees has been added to the figure legends. 

- Code and data sharing: to increase replication of the study by other researchers, authors should consider to share their data and code with the community.

- The authors thank the reviewer for this comment, and have added all code to the Github, and indicated this at the end of Methods.

---

## [Decision Letter · Decision Letter 1]

18 Aug 2022

PONE-D-21-34503R1Predicting Plasmodium falciparum infection status in blood using a multiplexed bead-based antigen detection assay and machine learning approachesPLOS ONE

Dear Dr. Rogier,

Thank you for submitting your revised manuscript to PLOS ONE. Your revised manuscript has mostly addressed the reviewers concerns and suggestions. One reviewer has raised a few minor points around wording which I feel are worth incorporating into the manuscript as it will provide a more balanced interpretation of the data. I would appreciate if the 4 modifications outlined below and in the reviewers section can be incorporated into the final submission. Separately, I have provided the reviewer with higher quality figures that were not available from the Editorial Manager Website to address the reviewers request. 

Could these 4 comments/suggestions be addressed and a revised manuscript submitted. 1. In lines 339-346, it should be written more clearly that optimal accuracy might not have been achieved in this study, because this study did not attempt to do it so. To achieve optimal accuracy, other statistical and machine learning methods should have been used and compared with the conditional inference trees.

2. With respect to my point about cutoffs, I agree that we should aim for using more advanced analytical techniques. However, given that malaria (sero-)epidemiologists are well aware of cutoff-based methods, the paper benefits of explicitly suggesting the link between the cutoffs suggested by the conditional inference trees and those that can be derived from finite mixture models. This suggestion connects the study with the existing literature. This connection can be done by revising the paragraph in lines 339-346.

3. In the Material & Methods, it is important to state that it was assumed that there was no sample contamination in the data analysed.

4. Sorry for raising this comment at this stage, but it seems that dataset is imbalanced and this might affect the accuracy predicted by conditional inference trees. This should be briefly discussed by acknowledging that there are (machine learning) methods that could have been used to correct for that.

We look forward to receiving your revised manuscript.

Kind regards,

Danny W Wilson

Academic Editor

PLOS ONE

Journal Requirements:

Reviewers' comments:

Reviewer's Responses to Questions

**Comments to the Author**

1. If the authors have adequately addressed your comments raised in a previous round of review and you feel that this manuscript is now acceptable for publication, you may indicate that here to bypass the “Comments to the Author” section, enter your conflict of interest statement in the “Confidential to Editor” section, and submit your "Accept" recommendation.

Reviewer #1: All comments have been addressed

Reviewer #2: (No Response)

2. Is the manuscript technically sound, and do the data support the conclusions?

Reviewer #1: Yes

Reviewer #2: Yes

3. Has the statistical analysis been performed appropriately and rigorously? 

Reviewer #1: Yes

Reviewer #2: Yes

4. Have the authors made all data underlying the findings in their manuscript fully available?

Reviewer #1: Yes

Reviewer #2: No

5. Is the manuscript presented in an intelligible fashion and written in standard English?

Reviewer #1: Yes

Reviewer #2: Yes

6. Review Comments to the Author

Reviewer #1: (No Response)

Reviewer #2: I thank the authors for addressing my comments. The revised version of the manuscript improved substantially. However, I have still five minor comments:

1. Figures remain slightly blurred in the version that I have received. Therefore, I asked the editor or Editorial Office to follow-up on that.

2. In lines 339-346, it should be written more clearly that optimal accuracy might not have been achieved in this study, because this study did not attempt to do it so. To achieve optimal accuracy, other statistical and machine learning methods should have been used and compared with the conditional inference trees.

3. With respect to my point about cutoffs, I agree that we should aim for using more advanced analytical techniques. However, given that malaria (sero-)epidemiologists are well aware of cutoff-based methods, the paper benefits of explicitly suggesting the link between the cutoffs suggested by the conditional inference trees and those that can be derived from finite mixture models. This suggestion connects the study with the existing literature. This connection can be done by revising the paragraph in lines 339-346.

4. In the Material & Methods, it is important to state that it was assumed that there was no sample contamination in the data analysed.

5. Sorry for raising this comment at this stage, but it seems that dataset is imbalanced and this might affect the accuracy predicted by conditional inference trees. This should be briefly discussed by acknowledging that there are (machine learning) methods that could have been used to correct for that.

7. PLOS authors have the option to publish the peer review history of their article (what does this mean?). If published, this will include your full peer review and any attached files.

Reviewer #1: No

Reviewer #2: No

---

## [Author Response · Author response to Decision Letter 1]

24 Aug 2022

Schmedes, et al. 

PONE-D-21-34503R1

Response to Reviewers

All references have been cross-checked for accuracy. 

1. In lines 339-346, it should be written more clearly that optimal accuracy might not have been achieved in this study, because this study did not attempt to do it so. To achieve optimal accuracy, other statistical and machine learning methods should have been used and compared with the conditional inference trees.

- The authors are in agreement with the reviewer’s assessment, and as not every possible method was tested here (with the same dataset), we could not make a statement that the greatest accuracy was achieved. We have revised this section on Line 343 to include the statement: “As such, it could not be stated that this approach utilized here would produce optimal accuracy, and further investigation…” 

2. With respect to my point about cutoffs, I agree that we should aim for using more advanced analytical techniques. However, given that malaria (sero-)epidemiologists are well aware of cutoff-based methods, the paper benefits of explicitly suggesting the link between the cutoffs suggested by the conditional inference trees and those that can be derived from finite mixture models. This suggestion connects the study with the existing literature. This connection can be done by revising the paragraph in lines 339-346.

- The authors thank the reviewer for this suggestion, but do not agree with the parallels being drawn between the antigen detection data being presented in this current study and antibody detection data (what is utilized for malaria seroepidemiology). Statistical methods for dichotomization, including FMM, have been widely discussed in the antibody (almost unanimously IgG) literature, but this does not apply to levels of Plasmodium antigens in blood samples – a completely different biological phenomenon. Our group has casually investigated cutoffs for antigen data in previous studies (cited here), but there is no real precedent in the literature outside of our previous work. 

3. In the Material & Methods, it is important to state that it was assumed that there was no sample contamination in the data analysed.

- The authors agree with this statement, and have added text to Line 134: “For all laboratory data collected for analyses, it was assumed there was no sample contamination.”

4. Sorry for raising this comment at this stage, but it seems that dataset is imbalanced and this might affect the accuracy predicted by conditional inference trees. This should be briefly discussed by acknowledging that there are (machine learning) methods that could have been used to correct for that.

- For analyses presented here, datasets from separate studies were analyzed independent of one another, and predicted accuracies were presented for each unique study/analysis. So while the authors would agree that categories are not equally represented within each dataset, we provide datasets from different malaria transmission settings as examples of how these outputs may be interpreted from different endemic settings. The authors feel they currently describe the limitations of the current datasets in Discussion, but we have now added additional text to explicitly state that ML methods exist to help deal with imbalanced datasets on Line 340: “Future studies on larger datasets should address optimal statistical tests and machine learning models for infection status prediction, as well as employ methods to correct for dataset imbalance.”

---

## [Decision Letter · Decision Letter 2]

12 Sep 2022

Predicting Plasmodium falciparum infection status in blood using a multiplexed bead-based antigen detection assay and machine learning approaches

PONE-D-21-34503R2

Dear Dr. Rogier,

We’re pleased to inform you that your manuscript has been judged scientifically suitable for publication and will be formally accepted for publication once it meets all outstanding technical requirements.

Kind regards,

Danny W Wilson

Academic Editor

PLOS ONE

Reviewers' comments:

Reviewer's Responses to Questions

**Comments to the Author**

1. If the authors have adequately addressed your comments raised in a previous round of review and you feel that this manuscript is now acceptable for publication, you may indicate that here to bypass the “Comments to the Author” section, enter your conflict of interest statement in the “Confidential to Editor” section, and submit your "Accept" recommendation.

Reviewer #2: All comments have been addressed

2. Is the manuscript technically sound, and do the data support the conclusions?

Reviewer #2: Yes

3. Has the statistical analysis been performed appropriately and rigorously? 

Reviewer #2: Yes

4. Have the authors made all data underlying the findings in their manuscript fully available?

Reviewer #2: No

5. Is the manuscript presented in an intelligible fashion and written in standard English?

Reviewer #2: Yes

6. Review Comments to the Author

Reviewer #2: Thank for trying to accommodate my comments in the revised version. I am happy with the revisions done.

---

## [Editor Report · Acceptance letter]

20 Sep 2022

PONE-D-21-34503R2 

Predicting *Plasmodium falciparum* infection status in blood using a multiplexed bead-based antigen detection assay and machine learning approaches 

Dear Dr. Rogier:

I'm pleased to inform you that your manuscript has been deemed suitable for publication in PLOS ONE. Congratulations! Your manuscript is now with our production department. 

Kind regards, 

on behalf of

Dr. Danny W Wilson 

Academic Editor

PLOS ONE